# New Perspectives of Multiplex Mass Spectrometry Blood Protein Quantification on Microsamples in Biological Monitoring of Elderly Patients

**DOI:** 10.3390/ijms24086989

**Published:** 2023-04-10

**Authors:** Jérôme Vialaret, Margaux Vignon, Stéphanie Badiou, Gregory Baptista, Laura Fichter, Anne-Marie Dupuy, Aleksandra Maleska Maceski, Martin Fayolle, Mehdi Brousse, Jean-Paul Cristol, Claude Jeandel, Christophe Hirtz, Sylvain Lehmann

**Affiliations:** 1IRMB-PPC, INM, Montpellier University Hospital, INSERM, CNRS, University of Montpellier, 34295 Montpellier, France; 2Department of Biochemistry and Hormonology, Montpellier University Hospital, University of Montpellier, 191 Avenue du Doyen Giraud, 34295 Montpellier, France; 3Centre de Gérontologie Clinique Antonin-Balmès, Montpellier University Hospital, University of Montpellier, 39 Avenue Charles Flahault, 34090 Montpellier, France

**Keywords:** microsampling, clinical chemistry, DBS, LC-MS

## Abstract

Blood microsampling combined with large panels of clinically relevant tests are of major interest for the development of home sampling and predictive medicine. The aim of the study was to demonstrate the practicality and medical utility of microsamples quantification using mass spectrometry (MS) in a clinical setting by comparing two types of microsamples for multiplex MS protein detection. In a clinical trial based on elderly population, we compared 2 µL of plasma to dried blood spot (DBS) with a clinical quantitative multiplex MS approach. The analysis of the microsamples allowed the quantification of 62 proteins with satisfactory analytical performances. A total of 48 proteins were significantly correlated between microsampling plasma and DBS (*p* < 0.0001). The quantification of 62 blood proteins allowed us to stratify patients according to their pathophysiological status. Apolipoproteins D and E were the best biomarker link to IADL (instrumental activities of daily living) score in microsampling plasma as well as in DBS. It is, thus, possible to detect multiple blood proteins from micro-samples in compliance with clinical requirements and this allows, for example, to monitor the nutritional or inflammatory status of patients. The implementation of this type of analysis opens new perspectives in the field of diagnosis, monitoring and risk assessment for personalized medicine approaches.

## 1. Introduction

To improve public health and patient care, there are many initiatives to provide non-invasive and cost-effective blood biological testing. One major element in this field is represented by microsampling approaches [1]. They include the well-known “dried blood spot” (DBS) and a multitude of devices that collect whole blood or plasma samples, with volumes ranging from two to fifty microliters on average [2]. The demand is high for these new micro sampling techniques because they have many advantages, including the fact that they use less invasive capillary sampling, can be performed outside the hospital environment (at home, etc.) and can be easily combined with point-of-care approaches. After drying of the DBS, delivery by post is possible as the stability of the proteins is demonstrated over several days [3]. Nowadays, medical deserts are present, especially in rural areas, making it difficult to access health facilities and the possibility of taking a blood sample [4]. Medical staff are stretched and waiting times are thus increased. In addition, given the aging population and increased life expectancy, the follow-up of chronic diseases drives a large part of analyses performed, with the need for periodic follow-up [5]. The number of analyses has, therefore, increased in recent years. Finally, with the advent of personalized medicine, the determination of markers is also part of the routine examination, further increasing the number of blood samples needed [6]. Being more prone to blood sampling, elderly patients may also present hematomas, especially when antiplatelet or anticoagulant therapy is in use [7]. Finally, with dehydration, elderly patients present a greater venous fragility, complicating the feasibility of standard techniques [8].

The DBS and microsample approach would allow patients to take their own samples and send them to the specialized laboratory since they do not need to be stored at −80 or −20 °C before analysis, unlike standard samples [9,10]. The capillary sampling system and DBS are commonly used in clinical analyses such as for blood glucose [11], HbA1c [12] or neonatal screening [13].

On the other hand, there is a strong trend to generate large panels of protein and metabolic “biomarkers” (>50) with the idea of defining biological profiles with the help of artificial intelligence and with the aim of predictive medicine [14]. In addition to the classical determination of the inflammatory and nutritional status with albumin, transferring and C reactive protein, other proteins provide information on immune metabolism, which is sometimes involved in cancer mechanisms, lipid metabolism or the haemostatic mechanism. These same proteins have implications in certain well-known pathologies such as diabetes (Retinol-binding protein 4 [15]) or cardiovascular diseases (Apolipoprotein(a) [16], Beta-Ala-His dipeptidase [17] and Apolipoprotein M [18]), thus allowing a broader analysis of the systemic health status of patients. The combination of micro-sampling and multiplex testing is of great interest in this context [19].

The challenge is to reconcile the analysis of a very small volume of biological samples with many clinically relevant tests. This requires technologies capable of providing multiple measurements with microliter samples. Ultrasensitive immunodetection approaches using a wide range of innovative nanoscale, microfluidic, PCR-based, or digital technologies [20] are capable, in their research format, of achieving such a feat. However, there is a huge analytical, medical and financial gap to move from the so-called “RUO” (research only) to “IVD” (in vitro diagnosis) tests. Thus, to our knowledge, virtually none of these new ultrasensitive technologies are available in clinical routine and certainly not in their multiplex format.

Mass spectrometry (MS), a reference analytical technique that measures the mass-to-charge ratio (*m*/*z*) of one or more molecules, is already used in some clinical laboratories for the quantification of non-protein markers such as small molecules and xenobiotics [21]. Until recently, MS was not adapted to protein biomarkers, which are among the most clinically relevant to explore metabolic, hepatic, renal, immune or inflammatory disorders. In comparison to commercially available immunoassay, MS methodology requires expensive instruments, expertise and time to develop and validate the analytical method. Despite this, MS methodology is promising if more specificity is required (e.g., proteoforms quantification) or in multiplex biomarker analysis.

Only recently have several groups [22,23], including ours [24], developed the multiplex clinical MS detection of the main blood protein biomarkers in different fluids [25]. This work was performed mainly on retrospective samples and was often not related to clinical practice. Demonstration of the practicality and medical utility of these approaches in a clinical setting was therefore lacking.

In this context, we decided to set up a clinical trial on elderly to compare a large series of biological tests carried out from micro-sampling procedures, including plasma microsamples and DBS. These patients would benefit most from non-invasive blood sampling that can even be performed at home. This type of trial is necessary to provide evidence-based medicine. Our main objective was the quantification of 62 protein biomarkers. As secondary objective, we aim to correlate quantification of the protein biomarker panel to the clinical indexes for the detection of frailty [26], which is a major problem in the follow-up of the elderly [27].

## 2. Results

### 2.1. Participants

Recruitment from two geriatric units allowed the inclusion of 46 patients with sufficient sampling to perform all measurements necessary to evaluate the various study outcomes. The characteristics of the population were appropriate for the study, with the clinical analytes of interest spanning the range of normal to pathological values (Table 1). This, therefore, allowed us to calculate clinical concordances between the different approaches. Instrumental activities of daily living (IADL) score, body mass index (BMI) and mini-mental state examination (MMSE) scores were also collected from a significant number of patients, to relate the biological results to clinical and frailty indicators.

### 2.2. Primary Outcome

From an analytical point of view, targeted LC-MRM analysis allowed the quantification of 62 proteins in two types of samples: plasma and DBS (Appendix A). These proteins belong to different groups such as apolipoproteins, complement proteins, or coagulation factors, and to different metabolic and physiological pathways. Twenty-six of these proteins are commonly used in clinical chemistry, as illustrated by the information on “LabTestOnline^®^” (Appendix A). The intra-assay median variability on plasma microsamples was equal to 3.5% (1.4–20.3) and to 4.0% (1.2–19.0) for DBS.

The inter-assay variation median was 4.2% (1.3–22.4) and was determined on 8 different QC samples (Appendix A). All the panel of proteins were detectable in the clinical range of the study.

The comparison between plasma microsampling and DBS was possible by calculating the correlation coefficient between the values of the 62 proteins obtained in the two situations (Table 2). A total of 48 proteins were significantly correlated between microsampling plasma and DBS with a *p* < 0.0001. Concerning C-reactive protein, Apolipoprotein A-IV and CD5_antigen-like protein, Pearson correlation coefficients were higher than 0.9. The lowest correlation was allocated to hemoglobin, which is associated with red blood cells that are present only in whole blood (DBS).

### 2.3. Secondary Outcomes

The individual values of these proteins allowed us to stratify patients according to their physiological or pathological status. IADL was used to link biological status and clinical scales related to frailty in the elderly. IADL score was highly correlated to ADL (activities of daily living), MMSE, TMM (mini motor test), walking speed with a *p*-value below 0.05. IADL scale was stratified between low (0 to 4) (*n* = 23) and high score (5 to 10) (*n* = 15) to separate patients in two groups. Patients managing most of their daily living independently or with supervision (high score), and patients having a significant impairment in their ability to perform these activities and may require more significant support or care were statistically compared.

ApoD was the best biomarker link to IADL score in microsampling plasma (*p* = 0.0061) and as well as in DBS (*p* = 0.0029). ApoE was the second-best biomarker (for plasma *p* = 0.0552 and for DBS *p* = 0.013, Figure 1). Alpha-2-macroglobulin in plasma microsampling (*p* = 0.0074) and insulin-like_growth_factor_binding_protein_acid_labile_subunit in DBS (*p* = 0.028) were also linked to IADL score. No other proteins were statistically different between high IADL and low IADL groups (Appendix A).

## 3. Discussion

In this study comparing MS biochemical analyses on two types of microsamples, plasma microsampling and DBS, we sought to determine whether the latter could provide analytical and clinically relevant information in the field. We initially focused on the MS quantification of a large panel of proteins. The first observation was the quantifiable proteins between the two different samples. In the plasma sample, red blood cells, white blood cells and platelets were removed, which had an impact on detectable protein biomarkers.

From plasma microsampling and DBS, we were able to quantify 62 proteins including five clinically relevant analytes in a geriatric population that often had significant venous fragility and risk factors for frailty: albumin, prealbumin, C-reactive protein, Alpha-1-acid glycoprotein 1 and transferrin. Regular monitoring of biomarkers, and especially nutritional and inflammatory ones, is, therefore, of interest in this population, especially if this monitoring is noninvasive, uses a small amount of blood and can be performed in a nonmedical environment. Analytical performances of the assays using the microsamples were satisfactory. This confirmed that clinical MS proteomics on microsamples reached a level of quality compatible with IVD use.

What made our multiplex MS approach interesting and unique was its ability to measure a large number of proteins of clinical interest from a minimal sample volume, and with good analytical performance. The quantification of multiple blood proteins existed for more than 10 years using MRM assays similar to ours [28,29] or after immunopurification of proteins or peptides after digestion (an interesting approach called SISCAPA [30]). Immuno-purification prior to MS allows better sensitivity but requires multiple antibodies, thus increasing cost and adding biases based on the fact that antibodies do not capture all isoforms of a given protein. Adaptation of multiplex assays to DBS was already implemented [21] but, to our knowledge, it was not yet evaluated under real-life conditions and tested for its clinical relevance. The challenge is indeed significant as it requires parallel measurements in patients with different microsampling methods. It is for this reason that we compared the correlation between protein levels in plasma microsamples and DBS. Most of them were significantly correlated. The protein with the lowest correlation was hemoglobin, which is associated with red blood cells that are present only in whole blood samples.

Our population was suitable for the study because the concentrations of the different analytes ranged from normal to pathological, thus allowing the clinical relevance of the different measurements to be assessed. With this in mind, we sought to identify potential protein biomarkers of IADL, an index assessing patients’ independence in instrumental activities of daily living in a geriatric population. Apolipoproteins E and D were significantly upregulated in patients with IADL below 5. Similar results were found in the literature, with increased apolipoprotein E levels in older adults [31]. Apolipoprotein E levels also appeared to be associated with both cognitive impairment [32] and Alzheimer’s disease and dementia [33]. In the same way, apolipoprotein D was significantly related to age and to Parkinson’s disease [34]. Moreover, in our study, the IADL scale was significantly correlated with the MMSE scale reflecting the cognitive status of the patients (*p* = 0.0104). This reinforces the idea that apolipoprotein D and E metabolism may reflect a cognitive disorder. Although the main objective of this research was not medical or economic, it can be noted that the cost of one MS run is much lower than the cumulative cost of standard analyses (Appendix A). However, many regulatory and technical steps remain to be taken before deployment in medical laboratories.

This cohort, comprising elderly patients with diverse biological values, made it possible to validate both the analytical and clinical concordance between the two types of samples. Thus, the multiplex assay of a large panel of clinically relevant blood proteins can contribute to the personalized follow-up of patients. Due to artificial intelligence, we will eventually be able to establish individualized protein profiles and become part of personalized medicine.

This study had limitations. The main one was that the measurement of proteins from microsamples using MS was pooled into a small number of analytical batches. In routine laboratories, analyses are usually carried out as the samples arrive, rather than being performed daily as in standard assays. The results obtained in this study were from a single analysis center, which can induce an effect related to this laboratory.

## 4. Materials and Methods

### 4.1. Design and Setting

This was a monocentric, diagnostic equivalence trial comparing the analytical and clinical performance of micro sampling procedures. Patients were recruited at the “Antonin Balmès” Gerontology Center under the supervision of Pr C Jeandel. The multiplex MS analyses were performed in the ISO 9001 clinical proteomics platform under the supervision of Pr C Hirtz.

### 4.2. Study Population

Patients from the geriatric service (age ≥ 65 years and <100 years) were eligible if their management included a complete blood workup with exploration of their nutritional and inflammatory status. Patients consulted in a day hospital, for loss of autonomy or gait disorders, in the absence of an acute medical event. These relatively autonomous patients often had nutritional and inflammatory parameters within the normal range. Other patients were hospitalized for a geriatric short stay, suffering from an acute medical event, with a more pronounced frailty profile, with more impaired nutritional parameters and often elevated inflammatory markers. Patients were excluded if they had skin disorders that increased the risk of adverse effects from capillary blood sampling. Patients were enrolled within 24 h of being identified as meeting inclusion criteria. No data other than those related to the patients’ normal clinical care were collected (Table 1). In addition to sex and BMI, we collected Lawton’s IADL ranging from 0 to 8 [35] for most patients. No race/ethnicity information was collected for this trial.

### 4.3. Blood Collection

As part of their routine examination, patients received a conventional venipuncture to collect plasma (EDTA K2 BD Vacutainer^®^ 367864, Becton Dickinson, Franklin Lakes, NJ, USA). Tubes were handled according to our hospital’s standard ISO15189 procedure [36] and transported within twenty-four hours to the laboratory. Two to five capillary blood spots were also collected on a TFN-Specimen collection card included in a kit prepared by SpotToLab^®^ (Montpellier, France). The cards were dried for two hours at room temperature before being placed in an individual zipped plastic bag and transported within twenty-four hours at room temperature to the Clinical Proteomics Platform. Six mm diameter punches were then made using an automated DBS Puncher^®^ instrument (Perkin Elmer, Waltham, MA, USA). Each punch was placed in a 1.5 mL Eppendorf tube (Lobind, Eppendorf, Hamburg, Germany) and stored at −80 °C before use. To reduce the impact of blood diffusion on the cards, we excluded DBS spots with diameters less than 8 mm or greater than 14 mm from the analyses [37].

### 4.4. Mass Spectrometry (MS) Analysis of Microsamples

Sample preparation was automated on AssayMap BRAVO (Agilent Technologies, Santa Clara, CA, USA) to reduce preanalytical variability. Briefly, proteins from a 6 mm DBS card punch were extracted with ammonium bicarbonate (50 mM) while plasma microsamples (2 µL) were initially treated with denaturing buffer. Protein samples were reduced and alkylated, cleaned according to a proprietary protocol (WO/2020/234287) and digested with trypsin prior to LC-MS analysis, as shown in Figure 2 and fully described in Methods Sup.

Dried samples were resuspended in an acetonitrile/formic acid–water mixture (2.0/0.1/97.9%) spiked with reference peptides (Appendix A) and analyzed in duplicate. Between each measurement, a blank was measured to avoid carryover. LC separation using a reversed-phase column and a 48 min multistage gradient (Appendix A) is described in detail in Sup Methods. Peptide quantification was performed on a QqQ MS system (LCMS-8060, Shimadzu Corporation, Kyoto, Japan) based on the PeptiQuant™ Biomarker Assessment Kit (BAK-76) from CIL (Cambridge Isotope Laboratories, Tewksbury, MA, USA). Some additional peptides produced by PeptideSynthetics (Fareham, UK) were added to complete this panel as follows: ESDTSYVSL [^13^C6]K for C-reactive protein; AADDTWEPFASGK[^13^C6, ^15^N2] for prealbumin; L[^13^C6]VNEVTEFAK for albumin and, TEDTIFL [^13^C6]R; and WFYIASAFR[^13^C6, ^15^N4] for AAG (alpha-1-acid glycoprotein 1) (Appendix A). The heavy peptides allowed the generation of calibration curves and determined the LLOQ.

The Opensource Skyline^®^ 20.2 software was used to analyze the MRM data. Peak detection was performed automatically by the software and verified manually. Excel software was used to calculate the heavy/light ratio. Specific regression curves were used to calculate the concentrations of the five clinical analytes also measured by standard methods. The complete procedure can be performed in two days and in batches of 46 samples.

### 4.5. Outcomes

The primary outcome of the study was the correlation between plasma microsamples and DBS using MS quantification for the 62 proteins. The secondary outcome of interest was the relationship between biological status and clinical scales related to frailty in the elderly (IADL).

### 4.6. Statistical Analysis

Intra-assay variability was calculated on all peptides/proteins using duplicate measurements of plasma and DBS microsamples. Inter-assay variability was assessed on eight independent measurements of a normal reference plasma (Cryocheck^TM^, Cryopep, Montpellier, France) used as internal quality control. Statistical analyses were performed with MedCalc software (20.210). Comparison of the methods was performed using correlogram calculation with determination of Pearson correlation coefficient. The normal/pathological value classification was based on clinical thresholds defined in the routine laboratory. Comparison of samples was performed with a Mann–Whitney test. In order to analyze the proteins associated with the IADL, the study population was divided into two groups: the first group corresponded to patients with an IADL inferior to 5, the second group to those with an IADL equal or superior to 5.

### 4.7. Study Approval

The trial protocol was approved by the ethics committee “CPP Sud-Méditerranée IV” under reference number 2013-A00115-40. Recruitment was performed between October 2015 and March 2017. Follow-up continued until September 2018. All analyses in the published protocol and analysis plan were specified before the completion of patient recruitment and biologic testing. Written informed consent was obtained from patients. The trial was monitored by the Montpellier University Hospital acting as the sponsor.

## 5. Conclusions

Multiplex detection of blood proteins from microsamples, in accordance with IVD requirements, is feasible and can be considered as part of a comprehensive approach to performing biological analyses. One application of this technology is the assessment of nutritional status and inflammation in the elderly, an important public health problem. However, not all analytes achieve sufficient clinical performance due to analytical limitations (in terms of concentration range) or differences in the nature of the samples (plasma/ whole blood DBS microsamples). The implementation of this type of analysis opens new perspectives in the field of diagnosis, monitoring and risk assessment for personalized medicine approaches.

## Figures and Tables

**Figure 1 ijms-24-06989-f001:**
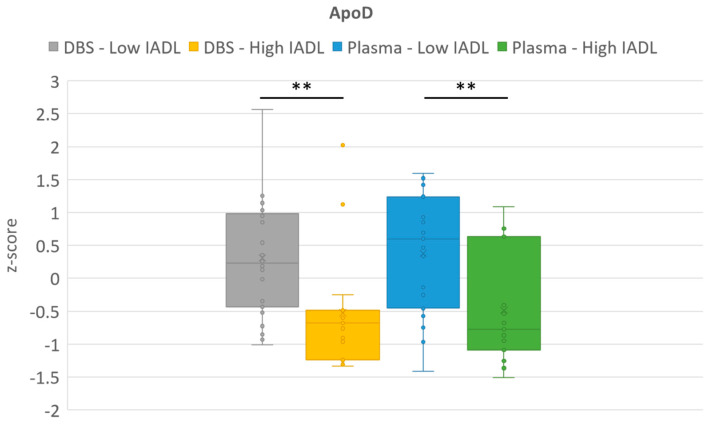
Boxplots of Apolipoprotein D (ApoD) and Apolipoprotein E (ApoE) protein levels as a function of the instrumental activities of daily living (IADL) index. Comparisons were made for both dried blood samples (DBS) and plasma microsamples. The Y-axis corresponds to the z-scores obtained for each of the proteins. Mann–Whitney *p*-values: * *p* < 0.05, ** *p* < 0.01. The circles correspond to individual values and the cross to the mean of the group.

**Figure 2 ijms-24-06989-f002:**
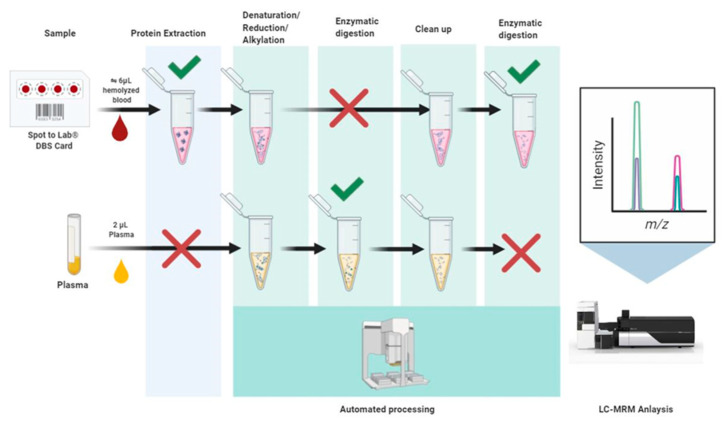
Generic workflow from plasma and DBS samples to LC-MS analysis. Generic representation of the sample preparation workflow before LC-MS analysis. After transferring to a 96-deep-well plate, multi-step sample preparations were performed on an AssayMap Bravo (Agilent Technologies, Lexington, KY, USA). A bottom-up approach was used to perform protein quantification by MS. To obtain optimal results, protein samples were extracted/denaturated/reduced and alkylated before the protein digestion with trypsin. Clean samples were then injected on the LC-MS instrument.

**Table 1 ijms-24-06989-t001:** Demographic and clinical characteristics of the population. Minimum, maximum and median values observed in the population were reported, as well as the 25th and 75th percentiles (25–75 P).

	N	Minimum	Maximum	Median	25–75 P
Age	46	69.8	95.3	84	80.3–87.7
Sex	46	15 males	31 females		
BMI	43	16.2	42.2	25.5	22.1–29.2
IADL	38	0	8	3	1.0–6.0
MMSE	30	3	29	23.5	19.25–25.75

Abbreviations: Instrumental Activities of Daily Living Scale (IADL), Body Mass Index (BMI), Mini-Mental State Examination (MMSE).

**Table 2 ijms-24-06989-t002:** Protein correlation of plasma microsampling and DBS. Significant *p*-value are underlying. Red color indicated a high Pearson correlation coefficient and a green color a low Pearson correlation coefficient.

	Pearson Correlation Coefficient	*p*-Value
C-reactive protein	0.984	*p* < 0.0001
Apolipoprotein A-IV	0.931	*p* < 0.0001
CD5_antigen-like	0.919	*p* < 0.0001
Alpha-1-antichymotrypsin	0.874	*p* < 0.0001
Lipopolysaccharide-binding_protein	0.873	*p* < 0.0001
Apolipoprotein(a)	0.872	*p* < 0.0001
Haptoglobin	0.849	*p* < 0.0001
Complement_factor_I	0.845	*p* < 0.0001
Beta-Ala-His_dipeptidase	0.844	*p* < 0.0001
Beta-2-microglobulin	0.83	*p* < 0.0001
Neuropilin-2	0.822	*p* < 0.0001
Serum_amyloid_A-4_protein	0.816	*p* < 0.0001
Retinol-binding protein 4	0.81	*p* < 0.0001
Apolipoprotein M	0.794	*p* < 0.0001
Complement_component_C9	0.781	*p* < 0.0001
Protein_AMBP	0.781	*p* < 0.0001
Heparin_cofactor_2	0.779	*p* < 0.0001
Alpha-2-macroglobulin	0.764	*p* < 0.0001
Prealbumin	0.76	*p* < 0.0001
Complement_C2	0.758	*p* < 0.0001
Insulin-like_growth_factor_binding_protein_acid_labile_subunit	0.755	*p* < 0.0001
Complement_C4-B	0.754	*p* < 0.0001
Coagulation_factor_XII	0.747	*p* < 0.0001
Apolipoprotein B-100	0.738	*p* < 0.0001
Complement_factor_D	0.73	*p* < 0.0001
Apolipoprotein C-II	0.723	*p* < 0.0001
Beta-2-glycoprotein 1	0.718	*p* < 0.0001
Transferrin	0.708	*p* < 0.0001
Coagulation_factor_X	0.707	*p* < 0.0001
Pigment epithelium-derived factor	0.706	*p* < 0.0001
C4b-binding_protein_alpha_chain	0.7	*p* < 0.0001
Alpha-1-acid glycoprotein 1	0.691	*p* < 0.0001
Apolipoprotein E	0.689	*p* < 0.0001
Complement_C3	0.687	*p* < 0.0001
Vitamin_K-dependent_protein_S	0.684	*p* < 0.0001
Lumican	0.674	*p* < 0.0001
Fibulin-1	0.668	*p* < 0.0001
Gelsolin	0.657	*p* < 0.0001
Alpha-2-HS-glycoprotein	0.651	*p* < 0.0001
Cholinesterase	0.651	*p* < 0.0001
Afamin	0.649	*p* < 0.0001
Apolipoprotein D	0.648	*p* < 0.0001
Fibrinogen_alpha_chain	0.63	*p* < 0.0001
sp|P00751|CFAB	0.626	*p* < 0.0001
Inter-alpha-trypsin_inhibitor_heavy_chain_H2	0.614	*p* < 0.0001
Prothrombin	0.563	*p* < 0.0001
Clusterin	0.56	*p* = 0.0001
Apolipoprotein A-I	0.545	*p* = 0.0001
Complement_C1q_subcomponent_subunit_C	0.53	*p* = 0.0002
Insulin-like_growth_factor-binding_protein_3	0.515	*p* = 0.0003
Hemopexin	0.498	*p* = 0.0004
Alpha-1B-glycoprotein	0.468	*p* = 0.0010
Thyroxine-binding_globulin	0.447	*p* = 0.0018
Complement_C1q_subcomponent_subunit_B	0.446	*p* = 0.0019
Vitamin_D-binding_protein	0.442	*p* = 0.0021
Complement_C5	0.434	*p* = 0.0026
Complement_C1r_subcomponent	0.424	*p* = 0.0033
Corticosteroid-binding_globulin	0.417	*p* = 0.0040
Alpha-2-antiplasmin	0.401	*p* = 0.0058
Albumin	0.319	*p* = 0.0307
Antithrombin-III	0.291	*p* = 0.0495
Complement_component_C8_beta_chain	0.258	*p* = 0.0834
Hemoglobin_subunit_alpha	−0.07	*p* = 0.6420

## Data Availability

The data presented in this study are available on request from the corresponding author. The data are not publicly available due to intellectual property concerns.

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
