# Peer review of "New Perspectives of Multiplex Mass Spectrometry Blood Protein Quantification on Microsamples in Biological Monitoring of Elderly Patients"

_ijms, 2023, doi:10.3390/ijms24086989_

Round 1

Reviewer 1 Report

The manuscript is about multiplex detection of plasma proteins using microsamples. The manuscript is well-written and easy to read, and the analytical methods seem to be properly developed. I have a few concerns that should be addressed before the manuscript can be accepted for publication:

  1. I see that the advantage of the authors' method is that sampling can be performed at home, but as the method requires analytical instruments, the samples should be measured in hospitals. Did the authors perform any study on the stability of samples? It can be an issue in real-life applications.

  2. In Fig. 1: the x-axis is missing, and the authors should correct it. The bigger problem is that two points are simply not enough to draw conclusions. The authors should either use more points or not use it to draw conclusions.

  3. The authors should explain why the protein correlation of plasma microsampling and DBS differs significantly in many cases.

Author Response

Point 1: I see that the advantage of the authors' method is that sampling can be performed at home, but as the method requires analytical instruments, the samples should be measured in hospitals. Did the authors perform any study on the stability of samples? It can be an issue in real-life applications.

Response 1: The advantage of this method is to allow minimal invasive sampling technic and to allow patients to take a sample on their own or at home with paramedical assistance. But in fact, this method requires analytical instruments that are only available in the laboratory, requiring the sample to be sent. To avoid confusions, the words “home testing” were replace by “home sampling” (line 17).

In this paper we did not investigate the stability of DBS samples over time but, based on the literature (Chambers AG, Percy AJ, Yang J, Camenzind AG, Borchers CH. Multiplexed Quantitation of Endogenous Proteins in Dried Blood Spots by Multiple Reaction Monitoring - Mass Spectrometry. Mol Cell Proteomics MCP. March 2013;12(3):781), the authors demonstrate that DBS can be stored for up to 5 to 10 days, even at high temperatures. Sentence was added in the introduction “After drying of the DBS, delivery by post is possible as the stability of the proteins has been demonstrated over several days” line 41.

In this study, DBS were “transported within twenty-four hours at room temperature”, punched and “Each punch was placed in a 1.5 ml Eppendorf tube (Lobind, Eppendorf) and stored at -80°C before use”.

Point 2: In Fig. 1: the x-axis is missing, and the authors should correct it. The bigger problem is that two points are simply not enough to draw conclusions. The authors should either use more points or not use it to draw conclusions.

Response 2: Thank you for your comment. We have reworked Figure 1 by adding the legend to the x-axis (line 153 and 154). We have also changed the format to a boxplot, which allows us to integrate all the values (points) as well as the error bars. We hope that this will help justify our conclusions.

Point 3: The authors should explain why the protein correlation of plasma microsampling and DBS differs significantly in many cases.

Response 3: For the majority of proteins, with the exception of complement component C8 beta chain and haemoglobin, the correlation between plasma microsamples and DBS was significantly positive. This is our major concern in order to use biomarkers in both types of samples.

Despite this, the rewiever is right and we have clarified the context: “The first observation is the quantifiable proteins between the two different samples. In the plasma sample, red blood cells, white blood cells, and platelets have been removed which has an impact on detectable protein biomarkers.” line 164

Reviewer 2 Report

The manuscript entitled "New perspectives of multiplex mass spectrometry blood proteins quantification on microsamples in biological monitoring of elderly patients" by Vialaret et al outlines a clinical study employing mass spectrometry to quantify a set of proteins in plasma and dried blood spot samples. The aims of the study are clear and the methodology (supplementary information) is well described. However, there are multiple issues with the presentation of the results and conclusions of the study, and I cannot recommend publication without extensive rework and subsequent review.

Most glaringly, while some comparison of DBS and plasma datasets is offered in table 1, there is no actual discussion of protein quantification in relative or absolute terms between either the two IADL-based patient groupings, or any other groups. A single paragraph is offered on ApoD and ApoE quantification specifically, and supported by figure 1. Figure 1 is, however, poorly designed and does not allow readers to draw any firm conclusions. The choice of a line graph is inappropriate. The Z-score metric is meaningless outside the context of Skyline - can the authors relate it to e.g. fold change? What are the error bars? Was this calculated based on all 38 proteomic datasets where IADL scores were available? How many patients were present in each grouping? Did any other proteins exhibit significant expression differences between these two, or any other two groupings? Until a complete treatment of the data is presented, the merits of this manuscript are very hard to judge.

A minor, related point - the only conclusions I can find regarding the use of DBS versus plasma for sampling are that a. most proteins in the panel are generally detected at similar levels in both sample types, and b. sampling DBS allows the detection of more proteins due to the inclusion of haemoglobin. The latter point is a bit perplexing, as the presence of haemoglobin in the results is not automatically a benefit due to its abundance frequently impacting the detection of less abundant, but more diagnostically valuable peptides. Based on figure 1, the difference between the levels of ApoD and ApoE between the two patient groupings appears more pronounced in plasma than DBS (although whether that is statistically the case is impossible to say, see comments regarding figure 1 above). Can the authors comment on that?

Author Response

Point 1: Most glaringly, while some comparison of DBS and plasma datasets is offered in table 1, there is no actual discussion of protein quantification in relative or absolute terms between either the two IADL-based patient groupings, or any other groups. A single paragraph is offered on ApoD and ApoE quantification specifically, and supported by figure 1.

Response 1: We thank the reviewer for this comment. No significant differences were observed except for the two proteins ApoD and ApoE). A new supplementary Table (Table S4) was added to present all the comparison results (line 150).

As presented in the “Primary Outcome”, 62 proteins could be quantified in both samples (plasma microsamples and DBS) for all patients. However, the data are not included in the article. A Supplementary Table (S2) has been added to present biological results for the different proteins in each type of samples (line 118).

Point 2: Figure 1 is, however, poorly designed and does not allow readers to draw any firm conclusions. The choice of a line graph is inappropriate. The Z-score metric is meaningless outside the context of Skyline - can the authors relate it to e.g. fold change? What are the error bars? Was this calculated based on all 38 proteomic datasets where IADL scores were available? How many patients were present in each grouping? Did any other proteins exhibit significant expression differences between these two, or any other two groupings? Until a complete treatment of the data is presented, the merits of this manuscript are very hard to judge.

Response 2: Thank you for your questions. In the initial Figure 1, we agree, it is difficult to get a sense of the population used. Figure 1 has been redone and split into two figures for clarity and boxplots were used to avoid line graph as you recommend. We now relate it to fold change and add the errors bar. This calculation was based on the 38 proteomic datasets for which IADL scores were available. Twenty-three patients were present in the low IADL group and fifteen patients were present in the high IADL group (line 141). No other proteins exhibit significant differences between these two groups (line 150). Data on all the other proteins were provide in a new supplementary table (Table S4).

Point 3: The only conclusions I can find regarding the use of DBS versus plasma for sampling are that a. most proteins in the panel are generally detected at similar levels in both sample types, and b. sampling DBS allows the detection of more proteins due to the inclusion of haemoglobin.

Response 3: Indeed, the two conclusions (a and b) are these because we focused on the fact that the results on the cohort were correlated between DBS and plasma microsamples (Table 2). This will allow similar population separations depending on the sample used. Both types of samples showed high levels of correlation for most of the proteins assayed, including those routinely used in the assessment of the inflammatory and nutritional status of patients. The protein showing the lowest level of correlation was haemoglobin. This is explained by the fact that haemoglobin is not present in plasma samples but only in DBS samples. To avoid misunderstanding, sentence “As a result, DBS allows the detection of more proteins compared to plasma microsamples” has been deleted (line 170), and next sentence was modified “From an analytical point of view, targeted LC-MRM analysis allowed the quantification of 62 proteins in both type of samples (Table S1 and S2).” (Line 117).

Point 4: The latter point is a bit perplexing, as the presence of haemoglobin in the results is not automatically a benefit due to its abundance frequently impacting the detection of less abundant, but more diagnostically valuable peptides.

Response 4: The reviewer is absolutely right. Lysis of whole blood releases large amounts of haemoglobin, which is deleterious in bottom-up approaches. With the targeted LC-MRM approach we were able to quantify in both fluids 62 proteins which is really positive in this methodology.

To remove your confusion, the challenge has been clarified “The first observation is the quantifiable proteins between the two different samples. In the plasma sample, red blood cells, white blood cells, and platelets have been removed which has an impact on detectable protein biomarkers.” line 164.

Point 5: Based on figure 1, the difference between the levels of ApoD and ApoE between the two patient groupings appears more pronounced in plasma than DBS (although whether that is statistically the case is impossible to say, see comments regarding figure 1 above). Can the authors comment on that?

Response 5: Patient groupings appears more pronounced in plasma than DBS in older version of Figure 1 but Figure 1 has been redrawn and the two proteins have been separated to avoid confusion. Indeed, the p-value statistics are not better when comparing the plasma samples, we will even find the opposite since the p-values are lower when comparing the DBS samples.

Due to the inter-individual variability, we don’t focus on the fold change and the comparison of them. A study on a larger number of patients would be necessary to refine the study of the fold change between plasma samples and DBS samples.

Reviewer 3 Report

A very interesting and comprehensive study, using a good methodological framework and statistically sound conclusions. However, several questions arise. In the introduction, the relevance and advantages of the proposed method are poorly indicated. The main focus is on markers of nutritional and inflammatory status. But the advantages of determining the level of other markers, which were used quite a lot, are not indicated. The remaining 60 markers were left without attention. How can their quantitative interpretation be used in diagnostics? What kind of states/diseases are these markers? Why were they chosen for this study? What are the potential possibilities of this multiplex approach when transferring it to the diagnosis of other states/diseases?

In my opinion, it is necessary to broadly develop an introduction to present more reasonable advantages of the proposed approach in comparison with the conventional ones.

Also a few notes on the text.

Line 17. How, according to the authors, can a home test be combined with mass spectrometry?

Line 28. Is this proposed approach in demand? What are the disadvantages of existing methods for assessing nutritional and inflammatory status?

Lines 58-59. A very dubious claim. How does this compare to the statement on Lines 142-143? Isn't mass spectrometry now used ubiquitously in clinics to evaluate markers? If no, the reasons for this situation are not given, for example, a comparison of the costs of MS and conventional routine analyzes.

Line 201. Sentence repeat.

Author Response

Point 1: In the introduction, the relevance and advantages of the proposed method are poorly indicated. The main focus is on markers of nutritional and inflammatory status. But the advantages of determining the level of other markers, which were used quite a lot, are not indicated. The remaining 60 markers were left without attention. How can their quantitative interpretation be used in diagnostics? What kind of states/diseases are these markers? Why were they chosen for this study? What are the potential possibilities of this multiplex approach when transferring it to the diagnosis of other states/diseases? In my opinion, it is necessary to broadly develop an introduction to present more reasonable advantages of the proposed approach in comparison with the conventional ones.

Response 1: Thank you very much for the points you have made. The introduction has been amended accordingly.

The remaining 60 biomarkers were more develop in the introduction as you suggest line 61 to 69.

“In addition to the classical determination of the inflammatory and nutritional status with albumin, transferring and C reactive protein, other proteins provide information on immune metabolism, which is sometimes involved in cancer mechanisms, lipid metabolism or the haemostatic mechanism. These same proteins have implications in certain well-known pathologies such as diabetes (Retinol-binding protein 4 [14]) or cardiovascular diseases (Apolipoprotein(a) [15], Beta-Ala-His dipeptidase [16] and Apolipoprotein M [17]), thus allowing a broader analysis of the systemic health status of patients. The combination of micro-sampling and multiplex testing is of great interest in this context [18]. »

Point 2: Line 17. How, according to the authors, can a home test be combined with mass spectrometry?

Response 2: The reviewer is absolutely right, this is a mistake on our part. “Home testing” was replaced by “home sampling”.

Point 3: Line 28. Is this proposed approach in demand? What are the disadvantages of existing methods for assessing nutritional and inflammatory status?

Response 3: The proposed approach is in demand and have been more developed line 43 to 56:

“Nowadays, medical deserts are present, especially in rural areas, making it difficult to access health facilities and the possibility of taking a blood sample [3]. Medical staff are stretched and waiting times are thus increased. In addition, given the aging population and increased life expectancy, the follow-up of chronic diseases drives a large part of analyses done, with the need for periodic follow-up [4]. The number of analyses has therefore increased in recent years. Finally, with the advent of personalized medicine, the determination of markers is also part of the routine examination, further increasing the number of blood samples needed [5]. Being more prone to blood sampling, elderly patients may also present hematomas, especially when antiplatelet or anticoagulant therapy is in use [8]. Finally, with dehydration, elderly patients present a greater venous fragility, complicating the feasibility of standard techniques [9].The DBS and microsample approach would allow patients to take their own samples and send them to the specialized laboratory since they do not need to be stored at -80 or -20°C before analysis, unlike standard samples [6,7].”

Point 4: Lines 58-59. A very dubious claim. How does this compare to the statement on Lines 142-143? Isn't mass spectrometry now used ubiquitously in clinics to evaluate markers? If no, the reasons for this situation are not given, for example, a comparison of the costs of MS and conventional routine analyzes.

Response 4: Line 58-59, we talk about the clinical use of mass spectrometry for non-protein markers as small molecules or xenobiotics (line 81).

Point 5: Line 201. Sentence repeat.

Response 5: Thank you, the repetition has been removed

Round 2

Reviewer 2 Report

The authors addressed all of my major concerns.

One remaining minor point - please ensure that the y-axis of both parts of figure 1 is labelled.

Author Response

Thanks to the reviewer for raising this point. The y-axis in Figure 1 has been labelled in the legend line 157 “ Figure 1. Boxplots of Apolipoprotein D (ApoD) and Apolipoprotein E (ApoE) protein levels as a function of the Instrumental Activities of Daily Living (IADL) index. Comparisons were made for both dried Blood Samples (DBS) and plasma microsamples. The y-axis corresponds to the z-scores obtained for each of the proteins. Mann-Whitney p –values: *p<0.05, **p<0.01. ”. We have also added the legend for the y-axis in Figure 1.
